# Sex Differences in Anderson–Fabry Cardiomyopathy: Clinical, Genetic, and Imaging Analysis in Women

**DOI:** 10.3390/genes14091804

**Published:** 2023-09-15

**Authors:** Denise Cristiana Faro, Valentina Losi, Margherita Stefania Rodolico, Elvira Mariateresa Torrisi, Paolo Colomba, Giovanni Duro, Ines Paola Monte

**Affiliations:** 1Department of Surgery and Medical-Surgical Specialties, University of Catania, Via Santa Sofia 78, 95123 Catania, Italy; 2Institute for Biomedical Research and Innovation, National Research Council (IRIB-CNR), Section of Catania, Via P. Gaifami 18, 95126 Catania, Italy; 3Institute for Biomedical Research and Innovation, National Research Council (IRIB-CNR), Via Ugo La Malfa 153, 90146 Palermo, Italy

**Keywords:** Anderson–Fabry disease, X-linked transmission, gender differences, cardiac hypertrophy, speckle tracking echocardiography

## Abstract

Anderson–Fabry Disease (AFD) is a rare, systemic lysosomal storage disease triggered by mutations in the GLA gene, leading to α-galactosidase A (α-Gal A) deficiency. The disease’s X-linked inheritance leads to more severe, early-onset presentations in males, while females exhibit variable, often insidious, manifestations, notably impacting cardiac health. This study aims to examine gender-based AFD cardiac manifestations in correlation with the variant type: classical (CL), late-onset (LO), or variants of uncertain significance (VUS). We analyzed data from 72 AFD patients (53 females, 19 males) referred to the “G. Rodolico” University Hospital, employing enzyme activity measurements, genetic analysis, periodic lyso-Gb3 monitoring, comprehensive medical histories, and advanced cardiac imaging techniques. Statistical analysis was performed using SPSS version 26. Our AFD cohort, with an average age of 45 ± 16.1 years, comprised 12 individuals with hypertrophy (AFD-LVH) and 60 without (AFD-N). Women, representing about 75% of the subjects, were generally older than men (47.2 ± 16.2 vs. 38.8 ± 14.6, *p* = 0.046). In the female group, 17% had CL variants, 43.3% LO, and 39.6% had VUS, compared to 21.1%, 36.8%, and 31.6% in the male group, respectively. Females exhibited significantly higher α-Gal A values (median 7.9 vs. 1.8 nmol/mL/h, *p* < 0.001) and lower lyso-Gb3 levels (1.5 [IQR 1.1–1.7] vs. 1.9 [1.5–17.3] nmol/L, *p* = 0.02). Regarding the NYHA class distribution, 70% of women were in class I and 28% in class II, compared to 84% and 16% of men, respectively. Among women, 7.5% exhibited ventricular arrhythmias (10.5% in men), and 9.4% had atrial fibrillation (10.5% in men). Cardiac MRIs revealed fibrosis in 57% of examined women, compared to 87% of men. Even among patients without LVH, significant differences persisted in α-Gal A and lyso-Gb3 levels (*p* = 0.003 and 0.04), as well as LVMi (61.5 vs. 77.5 g/sqm, *p* = 0.008) and GLS values (−20% vs. −17%, *p* = 0.01). The analysis underscored older age, decreased lyso-Gb3 deposition, reduced hypertrophy, and lesser GLS compromise in females, suggesting later disease onset. Severe cardiac patterns were associated with classic variants, while more nuanced manifestations were noted in those with VUS. Early GLS impairment in males, irrespective of hypertrophy, emphasized the role of subclinical damage in AFD.

## 1. Introduction

Anderson–Fabry disease (AFD, OMIM #301500) is a rare, X-linked lysosomal storage disorder that arises from mutations in the *GLA* gene (Gene Entrez: 2717; NCBI reference sequence: NM_000169.3; OMIM #300644; Locus Reference Genomic record LRG_672), which is responsible for encoding the enzyme α-galactosidase A (α-Gal A, EC 3.2.1.22; Uniprot P06280). Located at Xq22 on the long arm of the X-chromosome, this enzyme’s primary role is to break down globotriaosylceramide (Gb3) within lysosomes. The severity of α-Gal A activity deficiency causes glycolipid accumulation, specifically Gb3 and lyso-Gb3, triggering multiorgan dysfunction, lifelong treatment complications, and an increased risk of premature death [1,2]. The disorder’s prevalence ranges from 1:8454 to 1:117,000 in men, regardless of ethnicity [2].

First identified in 1898 by dermatologists William Anderson and Johannes Fabry, the disorder’s association with α-Gal A enzyme deficiency was firmly established only in 1967 [2]. The advent of Next Generation Sequencing (NGS) technologies has significantly advanced our understanding of AFD’s genetic underpinnings. By facilitating comprehensive newborn and high-risk population screenings, these technologies have identified over 1000 *GLA* gene variants. These variants include classical- and non-classical-associated mutations, as well as Variants of Unknown Significance (VUS). Many of them are unique or ‘private’ (i.e., confined to one or a few families), and the frequency of de novo variants is under 10% [3,4].

The American College of Medical Genetics and Genomics has categorized these variants into five classes based on the likelihood of causing disease/phenotype [5,6].

Biochemically, AFD exhibits two distinct phenotypes: the classic and the non-classic or late-onset forms. The classic form is commonly found in male patients displaying nearly absent or very low α-Gal A enzyme levels (< 1%), leading to early symptom onset and marked accumulation of Gb3 in various cell types. This complex scenario correlates with an increased risk of multiorgan failure and premature mortality. In contrast, the non-classical form results from residual α-Gal A activity and is characterized by a subtler phenotype, with symptoms typically manifesting later in life and often primarily involving the heart [7].

Clinical variability is a distinctive feature of AFD, even within a single family, with significant intersex variability: the extensive allelic heterogeneity of mutations, coupled with variable disease severity and symptom onset, often makes accurate diagnosis challenging. The classical form patients usually present missense and nonsense variants, whereas non-classical patients with residual α-Gal A activity (between 3% and 25% of normal levels) frequently display missense mutations or *GLA* gene splicing site variants [5]. Furthermore, identified genetic polymorphisms may influence disease onset and severity, either exacerbating or alleviating disease manifestations [1].

AFD, a sex-linked genetic disorder, has variable and more subtle manifestations in females, which is often explained by lyonization leading to cellular mosaicism. Females with X-chromosome skew favoring the wild-type allele may present few symptoms, while those expressing the mutated allele may experience a severe disease akin to hemizygous males [8,9]. This results in a more nuanced and unpredictable phenotypic manifestation in women, potentially involving individual organs, notably the heart, with potential severity. Females often demonstrate vital organ involvement such as the kidneys, heart, or brain, albeit with a latency of about a decade compared to males. Diagnosis in these patients, even with normal enzymatic activity, is facilitated through genetic testing to identify the X-chromosome mutation.

The diagnosis of AFD involves a comprehensive clinical, biochemical, and genetic evaluation, underlining the complex nature of the disorder [1,10,11].

Patient identification strategies vary. Some programs employ enzyme activity tests from leukocytes or plasma, while others utilize dry blood spot testing for larger-scale screenings [12]. Enzymatic activity assessment is effective in male patients, exhibiting significantly lower levels than in healthy individuals. However, this method often falls short in females due to enzyme activity overlap with normal ranges. Consequently, some labs measure lyso-Gb3 levels for stratification or proceed to gene testing directly in all females [1,13]. Gene sequencing is pivotal for all suspected AFD patients, particularly before starting targeted therapy.

For molecular diagnosis, bi-directional (Sanger) sequencing of the *GLA* gene’s seven coding exons and exon-intron boundaries is the gold standard. For females, when Sanger sequencing does not detect a mutation, techniques like multiplex ligation-dependent probe amplification or quantitative polymerase chain reaction (PCR) are employed to spot large deletions or copy number variations [3,4]. High-throughput NGS, featuring various gene panels including *GLA*, is increasingly utilized for high-risk patient cohort screening, like those with unexplained left ventricular hypertrophy (LVH) or hypertrophic cardiomyopathy (HCM), thus identifying numerous *GLA* VUS, requiring expert evaluation to exclude non-disease-causing ones [1,14,15,16]. Furthermore, several *GLA* variants previously considered disease-causing have been reclassified as of uncertain significance or likely benign [3]. Therefore, *GLA* variants need a personalized assessment, necessitating further mutation investigation [14,15,16].

Over half of AFD patients manifest cardiac symptoms, a significant cause of disease-related mortality and reduced life expectancy. Clinically relevant heart disease typically emerges in adulthood—around the third or fourth decade for males and a decade later for females [17]. It frequently presents as progressive concentric LVH, leading to misdiagnoses due to its resemblance to other HCMs [11,18,19]. Early diastolic dysfunction and heart failure with preserved ejection fraction occur until the disease advances and systolic function is also compromised. The N215S variant is the most common cause of predominant cardiac involvement [20].

The build-up of Gb3 in cardiomyocytes, valve apparatus, conduction tissue, and vascular endothelium leads to myocyte hypertrophy and vacuolization due to intralysosomal Gb3 deposits. The spectrum of cardiac anomalies in Fabry cardiomyopathy extends beyond this, involving factors such as increased trophic factors, chronic inflammation, oxidative DNA and protein damage, resulting in contractile elements degradation, ATP synthesis reduction, cell death, and hypertrophy and interstitial remodeling through neurohormonal activation [21,22,23].

Other peculiar features include disproportionate papillary muscle hypertrophy [24] and minor valvular anomalies. Symptoms can include exertional dyspnea, anginal chest pain, palpitations, and syncope, often related to arrhythmias due to glycosphingolipid deposition and fibrosis [21,25]. In advanced stages, cardiomyopathy shows significant LVH, regional thinning, and fibrosis, mainly within the posterolateral LV segments, causing regional contractility impairment [26]. Cardiovascular complications contribute to 40% of deaths, with sudden cardiac death (SCD) accounting for 62% of cardiovascular-related deaths [19]. Non-sustained ventricular tachycardia, male sex, LVH, and fibrosis shown by late gadolinium enhancement (LGE) at cardiac magnetic resonance ( CMR)are linked to SCD in AFD [19,27].

Recent literature shows advanced echocardiographic techniques, like left-ventricular global longitudinal strain (LV-GLS) and mechanical dispersion (MD) by speckle tracking echocardiography (STE), provide early diagnostic and prognostic markers [28,29], correlating with fibrosis, ventricular arrhythmias, and SCD [30]. A study recently published by our group demonstrated MD as a reliable prognostic predictor due to its association with increased ventricular arrhythmias [31].

Cardiac involvement significantly impacts disease manifestation and patient outcomes. The sex-linked nature and variable phenotypic manifestation, particularly in cardiac involvement, highlight the need for further genetic understanding of AFD.

This study aims to examine the cardiac manifestations’ differences between men and women in a population of AFD patients from our Multidisciplinary Reference Center and analyze women’s cardiac phenotype according to the genetic variant type: classical (CL), late-onset (LO), or Variants of Uncertain Significance (VUS).

## 2. Materials and Methods

### 2.1. Study Design and Population

Our study, conducted at the “G. Rodolico” University Hospital in Catania, Italy, is a retrospective, observational analysis of AFD patients. Our patient cohort encompassed 72 individuals (53 women, 19 men) over 18 years old who had been diagnosed with AFD; they were followed from January 2014 through February 2023. We also included a control group, comprising 40 healthy individuals (26 men, 14 women), for comprehensive comparative analysis.

Diagnosis of AFD was genetically confirmed by assessing for pathogenic variants of the *GLA* gene using PCR on peripheral blood samples of consented participants. This assessment was performed alongside analyses of α-Gal A enzyme activity and lyso-Gb3 levels, which were carried out at IRIB-CNR laboratories, in Palermo, Italy.

*GLA* variants were classified as per the American College of Medical Genetics and Genomics guidelines into categories: pathogenic variants, likely pathogenic variants, VUS, likely benign variants, and benign variants. We then divided the pathogenic and likely pathogenic variants into “associated with classical AFD phenotype and with late-onset or non-classical phenotype”. In tables and figures, we decided to group variants with controversial interpretations, still debated between VUS and likely benign/benign polymorphism under a single category called “of conflicting interpretation of pathogenicity” as reported in the ClinVar portal (https://www.ncbi.nlm.nih.gov/clinvar/, accessed on 31 July 2023), or simply “VUS” in some cases (for sake of conciseness), awaiting a future characterization and definitive, unambiguous classification [5,32,33].

All symptoms and signs associated with genetically confirmed AFD were meticulously documented in our database. As part of the comprehensive evaluation at diagnosis, we conducted laboratory and instrumental tests to fully characterize the manifestation, extent, and organ involvement monitoring.

All study participants underwent an extensive clinical examination which included a cardiomyopathy-oriented medical history, a thorough evaluation of cardiovascular risk factors, and a detailed assessment of major comorbidities. This comprehensive clinical assessment involved an electrocardiogram (ECG), 2D-color Doppler transthoracic echocardiography (TTE), and dynamic-ECG (D-ECG). For patients with an uncertain familial disease history or those being evaluated for the first time with a high suspicion of AFD, we administered a detailed historical survey to detect any related symptoms or disorders potentially linked to AFD.

Depending upon the findings, patients without apparent symptoms or changes on imaging were scheduled for annual follow-up, whereas those displaying signs or symptoms of AFD were evaluated for therapy as per consensus guidelines and underwent more frequent follow-up visits based on specific organ involvement. Our final study selection included patients who had optimal imaging quality for speckle tracking analysis.

### 2.2. Echocardiographic Assessment

A comprehensive TTE examination was performed using a Vivid-E95 echocardiograph (GE, Horten, Norway) equipped with a multi-frequency probe. The procedures were undertaken by our laboratory standards as well as the EACVI/ASE guidelines [34,35,36,37]. Various parameters have been assessed, including diastolic interventricular septal thickness (IVS), posterior wall thickness (PW), body surface area-indexed left ventricular mass (LVMi), and systolic (ejection fraction by biplane Simpson method—EF) and diastolic function (left atrial volume index—LAVi; tricuspid regurgitation velocity—TR-V; E/e’ ratio). Advanced echocardiographic techniques, such as Tissue Doppler Imaging (TDI) and STE, were utilized to assess LV-GLS and MD.

Post-processing of the images was conducted using the semi-automatic software EchoPAC (ver. 2.02, GE HealthCare, Chicago, IL, USA). LV-GLS was analyzed from the apical views (chambers 3-4-2), at a frame rate of 60–70/s, based on the mean of three consecutive cardiac cycles. MD was computed automatically as the standard deviation of the time to peak negative strain across all segments of the left ventricle. Time to peak strain is described as the interval from the onset of the QRS complex on the ECG trace to the peak negative longitudinal strain throughout the cardiac cycle. All echocardiographic examinations were performed by two skilled operators (VL and DCF) and verified by a single supervisory expert (IPM) for quality assurance.

### 2.3. Assessment of Clinical Outcomes

In the event of reported symptoms such as heart palpitations, syncope, or any form of dyspnea, a D-ECG was recommended. Arrhythmic episodes were identified via D-ECG monitoring, provided by the patients, or device interrogation in the case of patients equipped with implantable cardioverter defibrillators (ICDs). We incorporated documented episodes of ventricular fibrillation (VF), sustained and non-sustained ventricular tachycardia (SVT, NSVT), atrial fibrillation (AF, whether paroxysmal, persistent, or permanent), and paroxysmal supraventricular tachycardia (PSVT) into our analysis.

Exertional dyspnea, when present, was evaluated during the anamnesis and classified by the New York Heart Association (NYHA) classification.

### 2.4. Cardiac Magnetic Resonance (CMR)

In patients with AFD exhibiting LVH, we recommended advancing the diagnostic evaluation with CMR imaging, both with and without contrast. This recommendation was made in the absence of contraindications and was dependent on the patient’s informed consent and compliance.

A specified subgroup, presenting known or suspected cardiac involvement (characterized by LVH at echo, hypertrophy of the papillary muscles, significant alterations in LV-GLS and/or MD, or symptoms suggestive of palpitations, syncope, or chest pain), and patients with severe multi-organ impairment who were currently under therapy or deemed as potential candidates for initiation of enzyme replacement therapy (ERT) or migalastat, underwent D-ECG and CMR imaging with examination of LGE.

CMR examinations were conducted either within our hospital or at external centers. Due to these diverse settings, the protocols, machines, and magnetic fields utilized were not standardized. Quantification methods for LGE varied between centers, which made standardizing quantitative data challenging. Therefore, to ensure a uniform approach, LGE was expressed based on the number of segments it involved, using the established 17-segment echocardiographic model as a reference.

### 2.5. Ethical Considerations

This study was conducted in accordance with the Declaration of Helsinki. Written informed consent was obtained from all subjects involved in the study.

### 2.6. Statistical Analysis

Data are expressed as mean ± standard deviation (SD) for continuous variables with normal distribution, as median and inter-quartile range (IQR) for data with no normal distribution (according to results of Kolmogorov–Smirnov tests), and as number and percentage for categorical ones. Data were compared with unpaired Student’s *t*-test or Mann–Whitney U test for continuous variables as appropriate based on the distribution (and ANOVA and Kruskal–Wallis tests for comparison of more than two groups) and chi-square and Fisher’s exact tests for non-continuous ones. We added a Spearman correlation analysis applied to lyso-Gb3 values. Statistical significance was defined for *p* < 0.05, two-tailed test. The software was IBM SPSS Statistics ver.26.

## 3. Results

### 3.1. Fabry Disease Patient Population Overview

Our study encompassed 112 patients: 72 with Fabry Disease (AFD tot), 12 exhibiting concurrent left ventricular hypertrophy (AFD-LVH), and 60 without hypertrophy (AFD-N); 40 healthy individuals (NH) formed the control group.

The average age of the AFD group was 45 ± 16.1 years. AFD-LVH patients, who likely represented a more advanced disease stage, had a higher mean age (60.7 ± 7.2 years), while the AFD-N group was younger at 41.8 ± 15.6 years. The NH group’s mean age was 44.1 ± 13.9 years, which was quite similar to the AFD tot group. The average follow-up was 39 ± 25.8 months, and the median follow-up was 44 [IQR 20–50.5] months.

In the AFD-LVH group, 58.3% were NYHA class II, 33.3% were class I, and 8.3% were class III. In the AFD-N group, 81.7% were asymptomatic (class I) and 18.3% were class II, with none in class III or IV.

Two patients from the AFD-LVH group passed away during follow-up. The first, a 64-year-old woman with a classic E341X variant, suffered a fatal cardiac arrest due to VF during her last hospitalization for acute heart failure. The second was a 59-year-old male with a D165H variant, with a history of AF, heart valve replacement, and kidney transplant, who died from neurological complications following a severe stroke.

A single AFD-LVH patient required ICD implantation due to a pathologic electrophysiologic study, one AFD-LVH patient required pacemaker implantation due to a complete atrioventricular block, and three others from the same group exhibited a short PR interval on ECG. D-ECG was performed on 12 patients (16.7%): 8 AFD-LVH (66.7%) and 4 AFD-N (6.7%). Ventricular arrhythmias were identified in six AFD total patients, five of whom were AFD-LVH (41.7%), and one was AFD-N. AF was present in seven patients, all from the AFD-LVH group (58.3%), and PSVT was found in seven patients, six of whom were AFD-LVH, underlining the association of these conditions with hypertrophy in AFD. In none of the patients within our sample were significant signs or symptoms of dysautonomia reported. For additional details, refer to Table 1 and Table 2.

### 3.2. Genetic Variants, Biochemical Parameters, and Therapy

In the AFD total group, 18.1% harbored a CL variant, 41.6% a LO, and 40.3% a VUS. The distribution of these variants exhibited variations between AFD-LVH and AFD-N patients. AFD-LVH patients mostly had CL variants (41.7%), followed by LO variants (33.3%). Conversely, the AFD-N group mostly harbored LO variants and VUS (each 43.3%), with only 13.4% carrying a CL variant. One case stood out in which a patient with a hypertrophic phenotype and supraventricular arrhythmias had both the MYH7 mutation associated with HCM and the D313Y *GLA* variant linked with AFD. For a more in-depth analysis, we divided the data on *GLA* gene variants found in two population groups. As seen in Table 3, Panel A contains detailed data on variants among the entire population who underwent genetic analysis within the study period. This group comprised individuals with suspected symptoms or signs of AFD, as well as asymptomatic family members from the same genealogical tree. Among these 173 individuals, 126 tested positive. However, not all these individuals agreed to comprehensive cardiac clinical screening at our center or continuous clinical follow-up. Panel B presents the variants found specifically in the 72 patients with AFD who underwent extensive cardiac evaluation.

In the general population undergoing screening, the most common variants were S126G (18.7%), A143T (18%), and M51I (11.8%), followed by G395A and F113L. In the 72 AFD patients’ subset, the most frequent variants were the same, albeit with some variations in the distribution percentages (S126G, M51I, and F113L were the top 3). Notably, the D313Y variant was present in about 10% of both the general population undergoing screening and the study population (see Figure 1 and Figure 2).

No significant differences were seen in the median α-Gal A activity levels between AFD-LVH and AFD-N groups. However, basal lyso-Gb3 levels during the first visit were significantly higher in the AFD-LVH group (median value: 9.2 nmol/L) compared to the AFD-N group (value: 1.5 nmol/L, *p* < 0.01), indicating a more extensive organ involvement due to lyso-Gb3 accumulation in the AFD-LVH group. We have also supplemented our analysis with Spearman correlation for lyso-Gb3, with a moderate correlation between lyso-Gb3 levels and LVMi (coefficient 0.34, *p* = 0.018), and with GLS (coefficient 0.34, *p* = 0.016).

### 3.3. Echocardiography

AFD-LVH patients exhibited significantly increased thickness of the IVSd and PWd compared to AFD-N patients (median IVSd 13 mm versus 8 mm, PWd 12 mm versus 8 mm in AFD-N). An intriguing finding was that the thickness of the posterior (or inferolateral) wall in AFD-N patients was less than in the control subjects (8 mm versus 8.9 mm), with a *p* value slightly above statistical significance (0.05). The presence of papillary muscle hypertrophy was identified via ultrasound in four AFD tot patients, three of whom were from the AFD-N group.

LVMi, LAVi, and the E/e’ were significantly higher in the AFD-LVH group in comparison to the other two groups. As for TR-V max, it was numerically higher in AFD patients, even if not significantly. No significant differences were observed among the three groups in EF values, which had a median of 65%.

LV-GLS analysis revealed a median value of −11.5% in the AFD-LVH group, which was significantly lower than in the other two groups (*p* < 0.001); in AFD-N it was lower than in the control group (−19% vs. −20%, although not significant). In the two AFD groups, the longitudinal strain was impaired in the basal lateral and inferolateral segments: despite the overall GLS value being within the normal range in AFD-N subjects, lower values were found in those areas typically affected by AFD. For further details, see Table 4.

### 3.4. Cardiac Magnetic Resonance

In our study, a subset of 14 patients (19.4%) from the AFD tot group underwent CMR; these were fewer than recommended due to constraints like allergies to contrast media, severe chronic kidney disease (CKD), claustrophobia, and patient non-compliance. This included six AFD-LVH patients (50%) and eight AFD-N patients (13.3%). CMR data aligned with echocardiography, demonstrating similar wall thickness, LVMi, and ejection fraction patterns. LGE, indicating fibrosis, was present in 83.3% of AFD-LVH and 50% of AFD-N patients. Segments involved in fibrosis had a median count of 3 [IQR 2–8.5] in AFD-LVH and 2 [IQR 1.5–4.5] in AFD-N.

### 3.5. Analysis of Sex Differences: A Closer Examination of Female Patients with AFD

Our extensive study of AFD patients presents a gender distribution of approximately 75% female and 25% male. This distribution was consistent in the two AFD-LVH and AFD-N subgroups.

Given the significant female representation, we conducted a comprehensive sub-analysis to better understand the clinical manifestation of AFD in women, firstly within the total AFD population and subsequently within the AFD-LVH and AFD-N subgroups (Table 5).

Within the total AFD group, out of 72 patients, 53 were women and 19 were men. In terms of age demographics, women presented with a higher average age than men (47.2 ± 16.2 years versus 38.8 ± 14.6 years), suggesting a later disease onset in women, which is an observation in line with the existing literature that posits women are on average about eight years older than men at disease onset.

Regarding the genetic variants’ spectrum, most women (43.4%) carried an LO variant, 17% had a CL variant, and 39.6% had a VUS. Among men, most had a VUS (42.1%), and then LO (36.9%) and CL variants (21%). The most prevalent variants were S126G and A143T (16% each) among men and S126G and M51I (18.9 and 17%, respectively) in women. The Lys240GlufsTer9 variant was found to be the most common classical variant (represented in 10% of males and 4% of females). Details about variant types and specific variants are provided in Table 3 and Figure 1 and Figure 2.

In our analysis, we identified a significant difference in α-Gal A levels (*p* < 0.001): the median value among women was 7.9 nmol/mL/h [IQR 4.7 to 11.6], which is significantly higher than the median in men of 1.8 nmol/mL/h [IQR 0.3 to 3.35]. Similar differences were observed in lyso-Gb3 levels (median 1.9 nmol/L [IQR 1.5–17.3] in men versus 1.5 nmol/L [IQR 1.1–1.7] in women. Regarding arrhythmic events, four women had ventricular arrhythmias and five exhibited AF, compared to two instances of each in men. Examining the distribution of the NYHA functional class, approximately 70% of women were in NYHA class I and 28% in class II (only one in class III), compared to 84% of men in class I and 16% in class II. Furthermore, a higher proportion of women (26.4%) were on ERT than men (21%). A comparable percentage of both groups was on migalastat, approximately 10–11%.

Echocardiography revealed that in the AFD-LVH group, women were more represented (9 women versus 3 men). While EF and the E/e’ ratio were similar in both genders; a significant sex-based difference in LV-GLS was observed. Men had lower values (−17% vs. −19%, *p* = 0.02), suggesting less cardiac damage compared to women. MD values were slightly lower in women (37 vs. 41 ms), although not statistically significant. Despite the small sample size in the CMR subgroup, we observed a significantly higher number of LGE positives and segments affected by LGE in males.

In Table 6, the AFD-LVH subgroup’s sex differences are analyzed more closely. α-Gal A levels were found to be significantly different, with women showing a median of 8.5 nmol/mL/h against a median of 0.3 nmol/mL/h in men. Differences were also noted in LVMi values (median 115 versus 227 g/sqm in men) and GLS and MD, although not statistically significant.

In the AFD-N subgroup, consisting of 44 female and 16 male subjects, a significant age difference was observed, with women showing a mean age of 44 ± 16 years versus 35.7 ± 12.9 years in men. Regarding variants, 47.7% of the women exhibited an LO variant (versus 31.3% in men), while 40.9% had a VUS (versus 37.5% in men). Sex-based significant differences in α-Gal A levels and lyso-Gb3 levels were also found. Approximately 20% of the women underwent ERT, compared to 12.5% of men, while 13.6% of women and 6.3% of men were treated with migalastat. There were also differences in the NYHA functional class distribution (76% of women in class I versus 93.8% of men and 24% of women in class II versus 6.3% of men). Echo parameters revealed significant differences in LVMi and GLS (−20% in women vs. −17%, indicating earlier subclinical damage in men even before LVH development).

Then, we contrasted key characteristics of female patients with and without LVH (see Table 7 and Figure 3 and Figure 4). Notably, a higher proportion of hypertrophic patients (40%) harbored a classic variant, while in non-hypertrophic patients, 47.7% and 40.9% had an LO variant and a VUS, respectively. Significant differences were also observed in LVMi, LV-GLS, MD, arrhythmias, lyso-Gb3, and age, with LVH patients being, on average, older (62.5 vs. 44 years). See Figure 5 for an example of echo images of an AFD-LVH female patient.

In Table 8, we present, separately for both genders, a comparison of echocardiographic parameters between AFD patients without hypertrophy (AFD-N) and the NH group. No significant differences emerged among women. However, in the male subgroup, the only significant difference pertained to LV-GLS, which was notably lower in AFD-N patients compared to the healthy controls.

### 3.6. Insights from Lyso-Gb3 Values among the Groups 

Levels of lyso-Gb3 (see Table 1) are significantly higher in patients with manifest cardiac involvement and cardiac hypertrophy, correlating with LVMi (median value 9.2 in AFD-LVH vs. 1.5 in AFD-N), irrespective of gender.

We have also supplemented our analysis with a Spearman correlation analysis for lyso-Gb3, noting a moderate correlation between lyso-Gb3 levels and LVMi (coefficient 0.34, *p* = 0.018), and with GLS (coefficient 0.34 *p* = 0.016).

Logistic regression analysis confirmed an association with LGE positivity (HR 1.8, 95% CI 1.16–2.89, *p* = 0.008). When comparing male and female subjects (Table 5), a notable difference emerged, with elevated values predominantly seen in males along with a significant difference in α-galactosidase levels. A correlation also emerged with LGE segments in both females (coeff. 0.44, *p* = 0.02) and males (coeff. 0.67, *p* = 0.049).

Within the AFD-N subgroup (males vs. females), this significant disparity is maintained (*p* = 0.04). However, it ceases to be significant in the AFD-LVH group, likely because overt hypertrophy represents an advanced phenotype where sex-based disparities diminish, apart from a later onset in women already discussed in the text (Table 6). In the AFD-N group, lyso-Gb3 continues to correlate with the number of LGE segments (coeff 0.46, *p* = 0.01), but its correlation with LVMi becomes weaker (0.17). No significant correlations emerged within the AFD-LVH group. Among females, a significant difference in lyso-Gb3 exists between the LVH and AFD-N groups, anticipated by disparities in LVMi and the differing trajectory of cardiac damage (Table 8).

A total of 26/72 AFD patients were under therapy (36%). ERT was given to 25% of AFD total patients, including 58.3% of AFD-LVH patients and 18.3% of AFD-N patients. Chaperone therapy with migalastat was administered to 11.1% of AFD total patients, including 8.3% of AFD-LVH and 11.7% of AFD-N patients. It is worth noting that during the follow-up, there was a crossover between the two treatment groups. Specifically, one patient transitioned from migalastat to ERT, due to both personal preference and his physician’s recommendation, citing suboptimal efficacy. Conversely, another patient switched from ERT to migalastat after being identified as carrying an “amenable” mutation.

### 3.7. Subanalyses According to Therapy and Lyso-Gb3 Values

We also tried to examine the differences between treated and untreated cohorts, as well as patients with ERT and those on migalastat. We observed a significant difference in baseline lyso-Gb3 values across all groups (*p* = 0.006) and between those receiving therapy versus those not receiving therapy (*p* = 0.002). In echocardiographic analyses, we noted significant disparities in LVMi both across the three groups and between the therapy and non-therapy cohorts. The combined therapy group showed elevated LVMi and wall thickness values. GLS values were marginally lower in the therapy group, more so in those on ERT, while MD was mildly reduced. Among patients receiving therapy, males exhibited significantly lower α-galactosidase levels than females (*p* = 0.003), and older age was noted in females. Median lyso-Gb3 levels were lower in males but with a wider distribution range (1.5 to 17.3, vs. 1.2–7.6 in females). Both GLS and MD were slightly more impaired in males, even when receiving treatment. For females, the sole significant difference between those on therapy and those not was in the LVMi (*p* = 0.02), while lyso-Gb3 was higher (although not significantly, *p* = 0.06) in patients who received an indication for therapy. Among males, no such significant differences were observed. Similar trends were noted among females with and without LVH; signs of a less impaired clinical profile in terms of enzymatic activity, lyso-Gb3, GLS, and MD were more pronounced in those not slated for therapy, which can be attributed to a milder phenotype. For additional details on these comparisons, please refer to Appendix A.

**Figure 3 genes-14-01804-f003:**
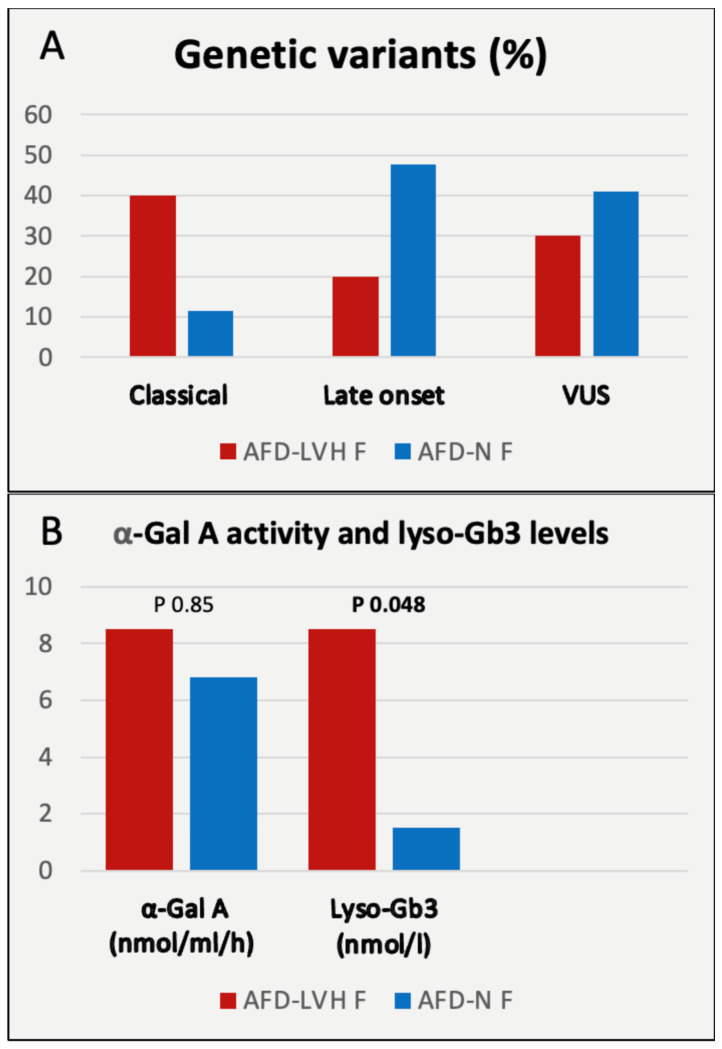
Comparison between females with AFD-LVH and AFD-N. Panel (**A**): genetic variants; panel (**B**): enzymatic activity and lyso-Gb3 levels.

**Figure 4 genes-14-01804-f004:**
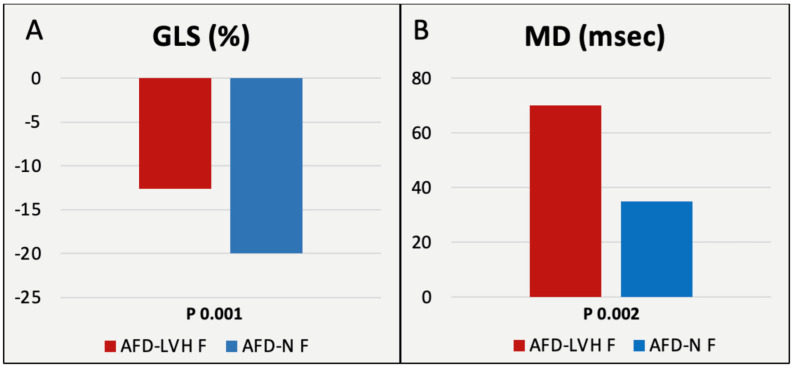
Graphical representation of the comparison between females with and without LVH: STE-derived parameters. Panel (**A**): global longitudinal strain (GLS); panel (**B**): mechanical dispersion (MD).

Moreover, we conducted an exploratory sub-analysis including only those patients for whom we had clinical parameters and advanced imaging prior to the initiation of therapy, as well as at least two follow-up evaluations; the first at 12–18 months and the second at the longest available follow-up. Advanced echography was performed at our center using standardized methodologies. Only 11 patients met these stringent criteria and were included in the sub-analysis, with a median maximum follow-up of four years. Due to the limited sample size and inherent heterogeneity of the patient groups—spanning varying phenotypes such as classical and non-classical, cardiac hypertrophy or not, and those on ERT or migalastat—no statistically significant differences emerged in the observed values. The trends, albeit not reaching statistical significance, appeared to suggest a decrease in lyso-Gb3 levels, an increase in α-galactosidase activity, a reduction in arrhythmia events, and an improvement in NYHA class. Conversely, the cardiac mass did not show beneficial effects from the treatment, nor did markers indicative of diastolic dysfunction. Parameters such as GLS and MD displayed a trend toward improvement over the follow-up period, though without achieving statistical significance. For further details, please refer to Appendix A.

### 3.8. Insights from Females and Males with Peculiar Manifestations

In analyzing patients with classic variants, common features were severe cardiac hypertrophy, reduced GLS, and increased MD. Specific cases among females include a patient (E341X) who had severe cardiac hypertrophy and passed away due to acute heart failure. Another patient (R220X) exhibited severe LVH (apical hypertrophy) and recurring AF (Figure 5).

**Figure 5 genes-14-01804-f005:**
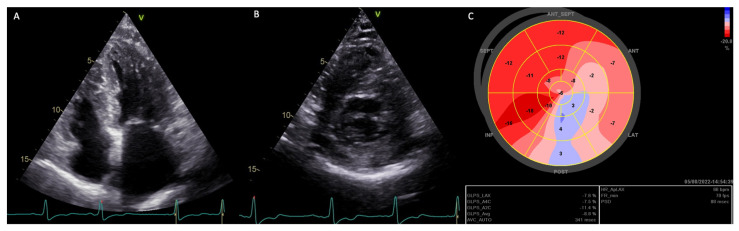
Echo images of a female AFD patient with LVH carrier of a pathogenic mutation linked to a classic form (R220X). Panel (**A**)—apical four chambers view; panel (**B**)—parasternal short axis view; panel (**C**)—global longitudinal strain (bull’s eye), mostly impaired in the inferolateral segments. LV-GLS average: −8.8%, frame/min 70, HR 66 bpm, MD 88 ms.

A patient with Lys240GlufsTer9 variant showed late-onset cardiac involvement despite high levels of α-galactosidase, with reduced strain, and episodes of NSVT and supra-ventricular arrhythmias; her son and a grandson, both carrying the same variants, developed multi-organ symptoms in their mid-20s with high lyso-Gb3 levels, papillary muscle hypertrophy and reduced GLS in the inferolateral segment.

Additionally, a 57-year-old female patient with the G85D variants initially presented with chest pain and slightly elevated troponin levels. She demonstrated mid-apical LVH and supra-ventricular arrhythmias within a multi-organ context. Interestingly, her 35-year-old daughter, carrying the same variants, retained a normal cardiac profile so far.

Patients with LO variants demonstrated distinct presentations. One patient (G395A) exhibited a negative cardiac phenotype from a clinical and echocardiographic viewpoint, but CMR indicated a spongy myocardium pattern at apical segments.

Patients carrying the D313Y variant displayed different degrees of cardiac involvement. One showed repetitive ventricular ectopies and NSVT runs in the presence of a normal ETT without hypertrophy, normal GLS, but slightly increased MD. Another presented mild hypertrophy with borderline diastolic dysfunction and reduced GLS, recurrent palpitation and syncope, and atypical chest pain. Another patient with the S126G variant showed borderline thickness and GLS values with minimal segmental strain alterations and underwent percutaneous revascularization for chronic coronary syndrome.

Among males with LVH, a notable case is a patient with the classic D165H variant, who passed away, and who demonstrated arrhythmic expressivity, with AF and ICD implantation for positive electrophysiological study, and fibrosis on CMR. Contrastingly, a male patient with the LO I91T variant had only mild LVH and no arrhythmias.

Another LO variant, F113L, manifested as marked cardiac hypertrophy. A female patient with this variant showed also multi-organ involvement and significant peripheral vasculopathy. A male patient with this variant exhibited massive septal hypertrophy, very low GLS, increased MD, and extensive fibrosis. He also underwent pacemaker implantation at 68 years of age due to AVB III. He also had runs of NSVT and AF episodes detected during D-ECG and pacemaker checks, indicating arrhythmic expressivity. His daughters had a mostly negative profile, except for one 40-year-old with papillary hypertrophy. These variants, I91T and F113L, were previously linked to a mild cardiac phenotype [38,39,40].

## 4. Discussion

Our investigation sheds light on the phenotypic discrepancies across various forms of Fabry disease, with a specific focus on sex-based manifestations. The clinical and instrumental findings revealed substantial heterogeneity. Echocardiography primarily unveils overt cardiac hypertrophy, allowing stratification into two clinically and prognostically distinct cohorts: AFD-LVH and AFD-N.

AFD-LVH patients demonstrated more severe symptoms and a higher propensity towards NYHA class II or III. ETT findings corroborated these clinical manifestations with elevated LVMi, higher wall thickness, and discernible diastolic dysfunction, evident in LAVi and E/e’ values. These abnormalities surpassed those in both AFD-N and control groups, further emphasizing the morphological and functional compromise in AFD-LVH.

The AFD-LVH group predominantly presented classic mutations, followed by late-onset ones. Our previous studies have established a correlation between elevated MD and ventricular arrhythmias, which is a trend consistent in hypertrophic and mildly elevated in non-hypertrophic AFD patients. Coupled with decreased GLS, this reinforces the importance of early cardiac involvement detection, to forestall fibrosis and arrhythmias.

In contrast, AFD-N patients displayed relatively normal ETT findings, albeit with reduced PW thickness than the NH group and lower GLS and MD values. These observations indicate an incipient phase of subclinical cardiac impairment, advocating for proactive monitoring given the potential early initiation of therapy.

As for CMR, in the AFD-LVH group, the basal and middle inferolateral segments presented LGE, aligning with the literature findings [41,42,43,44]. It is noteworthy that hypertrophy of papillary muscles, a potential AFD marker, even if non-specific, was observed in two patients [24]. One patient also exhibited fibro adipose tissue in the right ventricular wall, a feature common in arrhythmogenic right ventricular cardiomyopathy. In a patient with a variant associated with a late-onset form (G395A), the discovery of spongy myocardium at CMR, without LVH, yet not meeting criteria for myocardial non-compaction, is noteworthy. Although a case has been reported in the literature in a female with a LO variant (F113L) [45], this association has not been confirmed as subsequent extensive screening showed no additional AFD cases among patients with LV non-compaction [46]. LGE, indicative of advanced AFD stages and consequential macroscopic fibrosis, usually concentrates in the inferolateral basal wall of the left ventricle. Its persistence in post-replacement therapy signals an elevated risk of malignant arrhythmias and SCD, underscoring the critical prognostic role of LGE in AFD [19,47].

### 4.1. Sex Differences in AFD Cardiac Phenotype

Regarding the sex differences within the AFD-LVH and AFD-N subgroups, it is evident that women (comprising 3/4 of our population) exhibit different clinical manifestations. This group predominantly presented LO variants (43.4%) and only 17% CL variants. The average age was approximately 8 years higher in women than men, reflecting the disease’s later onset and corroborating literature findings [42,48,49]. Women exhibited significantly higher α-Gal A and lower lyso-Gb3 levels than men, aligning with previous reports showing that men typically present a more overt enzyme deficiency, while in female hemizygotes the activity of α-Gal A can result in the normal range, lyso-Gb3 is low at birth and increases gradually with age, likely with a correlation with symptomatology [48,50]. Additionally, women exhibited lower LVMi and less wall hypertrophy.

Remarkably, LV-GLS values in AFD-N were higher in women than men, underlining the early alteration of this parameter in subclinical disease stages; this is confirmed by Avanesov’s 2023 work showing sex-specific variance in LV-GLS and circumferential strain, with significantly lower values in male classic AFD patients compared to females [51]. These differences could be linked to increased myocardial sphingolipid storage, leading to impaired strain [52]; moreover, male patients exhibited higher LVMi and LGE, causing stiffer ventricles, while female patients often present without LVH, making early identification challenging.

Comparing AFD-N and NH, no significant differences emerged among women; there was only a trend toward slightly higher MD in AFD-N patients compared to controls. In contrast, in men, LV-GLS was significantly lower in AFD-N compared to NH, suggesting a reduction even before hypertrophy development. This could serve as a hypothesis-generating starting point for further evaluations regarding early diagnosis with promising implications for therapy.

As for CMR, we found a significant difference in the proportion of patients with fibrosis (LGE), which was higher in men. We did not analyze the difference according to LVH/no-LVH subgroups due to the small proportion of patients who underwent CMR (our wish is to do this soon), but we also found fibrosis in patients without hypertrophy (one out of three females and four out of five males). However, there is growing evidence that females especially may develop myocardial fibrosis before LVH occurs [22,44]. Our data contrasts with Niemann’s findings [41], where myocardial function loss and fibrosis in female patients with AFD did not necessarily require myocardial hypertrophy. In the female cohort, LGE occurred earlier than LVH and was detected in 23% without hypertrophy. For male patients, moderate LVH was required for fibrosis, whereas in women, fibrosis seems to progress regardless of hypertrophy, emphasizing the role of noninvasive assessment of fibrosis in females.

Differences were found among females between AFD-LVH and AFD-N groups in ventricular arrhythmias, LVMi, E/e’, LV-GLS, MD (corresponding to a significant difference in arrhythmias), and age, with AFD-LVH patients being older.

The most severe cardiologic manifestations were observed in women with classic variants (among AFD-LVH females, 40% had the CL variant, a significant difference compared to AFD-N with 11%), while those with VUS displayed more indistinct pictures. However, no significant gender differences were found for other echocardiographic parameters.

Positioning our work within the context of the most recent scientific literature on the subject, Azevedo et al. report that 40.2% of patients exhibited LVH and 67.1% had diastolic dysfunction. LVH, heart failure, and arrhythmias were more common in males and increased with age, typically manifesting in males over 30–40 years and females over 50 years. LGE occurred in 4.8% of patients without LVH and 52.5% with LVH, without gender differences. AF was found in 4.4%, NSVT in 8.9% (mainly males), and pacemaker implantation in 6.4%, occurring two decades earlier in males [53].

In a report from a Spanish multicentric registry of 97 females with AFD, variants associated with both classic and non-classic phenotypes were almost equally distributed (40.2%/53.6%). Females with classic variants exhibited higher frequencies of LVH (38.5%), dyspnea (11%), AF (2.2%), VT (4.4%), and LGE (2.2%), suggesting that females with pathogenic *GLA* variants are likely to develop symptoms regardless of the variant type [41].

In a study by El Sayed et al., LVH was observed in 50% of AFD patients (mainly in males with the non-classic form), with 66% undergoing CMR, and LGE was found in 39% overall (33% in classic men; 55% in non-classic men; 41% in classic women; 33% in non-classic women). MACE risk (combined endpoint cardiovascular death, heart failure hospitalization, sustained ventricular arrhythmias, and myocardial infarction) was influenced by sex and phenotype; it was the highest in men with classical AFD (11.0 per 1000 patient-years), intermediate in women with classic AFD and men with non-classic AFD, and lowest in women with non-classic AFD. Overall, the risk and timing of events differed by sex and AFD type, with women with classic and men with non-classic AFD experiencing events occurring approximately a decade later and with lower rates [42].

Appendix A provides an interesting comparison of our findings, related to the most significant (and comparable) parameters, to the historical classical cohorts from the Fabry Outcome Survey and from FAMOUS study (in relation to cardiac outcomes reported in published works), and also from recent works from El Sayed [40] and Avanesov [49], even though the focus of the latter study is predominantly on CMR imaging. Appendix A additionally juxtaposes our findings in the female patient cohort with those published by the Spanish Registry [39] and the study by Avanesov, both of which were previously discussed.

It should be clearly noted that only common and comparable parameters between the studies were set head-to-head for a cursory glance, taking into consideration the limitations due to differences in units of measurement, indexing methods, and measurement and reporting methodologies.

### 4.2. Genetics and Specific Variants’ Contributions

In our study, females displayed a higher percentage of LO variants than CL, which was also observed in males in our population, with specific frequent variants identified in both sexes. In females, the most frequent were S126G, M51I, and F113L; in males, S126G, A143T, D313Y, and F113L.

D313Y was present at about 11% in both sexes and is widespread in our region. The different pathogenetic variants of AFD are linked with particular clinical manifestations [54,55], such as D165H and R220X with typical AFD with LVH in both genders [56,57]. Variants like E341X and Lys240GlufsTer9, connected with classic AFD and LVH, were also noted in our patients. Concerning LO pathogenetic variants, our findings appear consistent with the existing literature, such as the F113L mutation’s constant association with cardiac damage, and other identified mutations in patients with a mild cardiac phenotype [38,39,40]. M51I has been correlated with atypical phenotypes, linked to cardiac involvement with symptoms like dyspnea and arrhythmias, as previously observed in an extended Italian family [58].

These observations add further layers to the complex genotype–phenotype relationships in AFD, highlighting the need for continued research to unravel the specific contributions of these variants to the disease’s clinical expression. Clinically, these pictures are typical and severe in forms linked to classic pathogenic variants in both males and females, with pronounced single-organ (in our case, cardiac) involvement even in LO variants. However, ascertaining genotype–phenotype associations in Fabry disease is challenging due to the disease’s rarity, marked allelic heterogeneity, variation in clinical expressivity, and lack of supporting published clinical data. Phenotype classification is still lacking for many *GLA* variants, and better comprehension of genotype–phenotype relationships is essential. Consideration must also be given to possible variability in the phenotype influenced by phenotype-modifying factors like genetic background, epigenetics, and environmental conditions [33]. The multispecialty Fabry disease genotype–phenotype workgroup created a novel Fabry phenotype consensus classification system but was limited to male phenotypes due to the multifactorial phenotypic variability in females with specific pathogenic *GLA* variants [5].

In our cases, the clinical pictures for VUS are more nuanced, with a lower percentage of LVH and classic phenotype, although some patients, especially women, show signs potentially indicative of AFD. The scientific community is divided over the interpretation of pathogenicity and causality for certain variants, with some studies categorizing them as VUS and others defining them as likely benign or even benign [5].

For example, two patients in the AFD-LVH group (D313Y) showed only cardiac involvement. One of them had a double mutation for HCM and AFD, blurring the line between sarcomeric HCM and Fabry contributions. Another patient with S126G presented a nuanced multi-organ picture with LVH, ischemic heart disease, AF, kidney transplant, and aspecific diffuse alterations, potentially influenced by external and environmental factors. A male patient with A143T displayed mainly nephrological and gastrointestinal features, but at CMR there was LGE in typical locations across four segments, along with regional GLS impairment, while a woman with S126G showed alterations detectable only through advanced techniques like CMR and GLS. In most patients affected by these controversial variants, we observed no specific cardiac pictures.

Variants initially labeled as VUS can be reclassified as either likely pathogenic/pathogenic or likely benign/benign, with the latter no longer regarded as AFD-causing. The A143T and D313Y variants are common in European newborn studies [59]; D313Y and S126G are described as strongly linked to AFD and contribute to the high enzyme activity of ≥3 5% of the normal range [60].

The variant p. (Ala143Thr) (A143T) is a particularly contentious case. Once considered pathogenic and leading to unnecessary treatments, an expert panel now regards it as likely benign, although the workgroup *GLA* variant phenotype classification consensus still views it as a VUS [33]. The debate around this ‘likely benign’ variant, often identified at a high frequency in screening programs, highlights the complexity of determining whether it should be classified as pathogenic [33,34].

The understanding of specific variants such as S126G and D313Y is complex and filled with contradictions in the literature. While some publications have claimed the S126G variant to be pathogenic when associated with a given haplotype, no convincing clinical evidence has been provided. The referenced expert opinion along with all major databases considers p. (Ser126Gly) as likely benign [5].

As for D313Y-p. (Asp313Tyr), it has been alternatively reported as a VUS or likely benign, and even referred to as polymorphism or non-pathogenic [57,61,62]. Contradictory evidence exists, with some studies affirming its pathogenic nature, showing late-onset milder phenotypes (only two females with LVH out of 17 cases, mostly neuro, nephro, and ocular involvement with a high a-GAL residual activity and normal lyso-Gb3 levels) [62]. Other research claims its non-pathogenicity is based on high allele frequency, normal plasma lyso-Gb3 levels, and lack of cardiac involvement at multiparametric imaging [51,63]. Functional studies indicate that p. (Asp313Tyr) is associated with 75% of normal α-Gal A activity (pseudo-deficiency), so three major databases (dbFGP, ClinVar, and LOVD) classify it as benign/likely benign [5].

Nevertheless, given remarkable clinical manifestations such as unexplained LVH in some patients, the exclusion of AFD has been cautioned despite normal lyso-Gb3 levels and VUS genotypes [62]. The significance of these variants remains unclear, and their relevance in clinical manifestations is still controversial, underscoring the need for continuous investigation and rigorous assessment to ascertain the pathogenicity and clinical relevance of these specific variants.

### 4.3. Fabry Cardiomyopathy in Women: Phenotype and Genotype

Contrary to common belief, females are not merely carriers of a defective *GLA* gene, but they can present with various manifestations of Fabry disease, ranging from the absence of symptoms or mild, late-onset phenotypes usually affecting only a few organs, to severe phenotypes, similar to the classic phenotype observed in male patients [1,48].

Regarding cardiac involvement, LVH is the most common manifestation, and it is more prevalent in males, but presents later in females by approximately 10 years. Arrhythmias, including permanent and paroxysmal AF, are relatively common in both sexes (14% and 4%, respectively). However, NSVT appears to have a higher incidence in males (8%) [48].

Both genders experience reduced quality of life, especially after their mid-30s with the diagnosis often delayed by about 16 years after symptom onset in women, who can experience significant multisystemic disease and should be monitored and treated accordingly [64,65].

Phenotypic variability in women primarily depends on the degree and direction of X-chromosome inactivation [49]. Some studies suggest a potential link between X-inactivation patterns and the clinical phenotype [8,9]. Depending on which X-chromosome is randomly ‘turned off’ or ‘turned on’, the expression of symptoms can vary across cells and organs within the same individual. However, classic measures of X-chromosome inactivation skewness may not fully capture disease manifestation: a study by Wagenhauser et al. showed that X-inactivation patterns alone do not reliably reflect the clinical phenotype of women when assessed in biomaterial not directly affected by the disease [66]. Additionally, allele-specific DNA methylation at the *GLA* gene promoter could influence the mutated allele’s expression levels, thereby impacting disease onset and outcome [67,68]. 

Thus, our understanding of the disease’s pathogenic mechanism is still incomplete. DNA methylation studies have been proposed as a means of understanding the complexity of Fabry disease, with potential implications for determining the appropriate timing for the initiation of therapy.

A 2022 study from the Fabry Registry showed agalsidase might slow heart muscle thickening and maintain normal kidney function in women with AFD [69], underscoring the importance of early detection and treatment to manage AFD effectively.

### 4.4. Impact of Therapy (ERT and Migalastat) in the Main Historical Fabry Cohorts, and Role of Lyso-GB3

Lyso-Gb3 serves as an increasingly critical biomarker in AFD, particularly in male patients with a classical phenotype, where it is elevated in plasma, offering high sensitivity and specificity [70]. This elevation is less pronounced in females and those with a later-onset phenotype. Beyond its diagnostic utility, lyso-Gb3 is instrumental in monitoring disease severity, genotype, and phenotype [15,71,72,73]. Initial studies suggest that ERT significantly reduces lyso-Gb3 levels in males with elevated baselines, although the clinical impact of this reduction remains uncertain [74,75]. The effect of migalastat, a chaperone therapy, on lyso-Gb3 levels is still inconclusive [74,76]. Further complicating its role, lyso-Gb3 may contribute directly to FD’s pathogenesis by promoting inflammatory processes and smooth muscle cell proliferation [32]. As it potentially correlates with lifetime exposure to the disease, it could serve as an efficacy marker for therapies, including ERT and migalastat, and is a useful and reliable screening tool [77]. Therefore, while its importance in FD management is growing, more robust, longitudinal studies are needed to fully validate lyso-Gb3’s utility in assessing treatment responses.

Regarding the follow-up analysis in our sample, limited by the number of patients we were able to include (based on the completeness and consistency of their records), from baseline to the one-year follow-up, and maximum available follow-up at a median of 4 years the median values tended to decrease, albeit without statistical significance. This lack of significance is likely attributable to the small and heterogeneous sample size, which included both ERT and migalastat treatments. From our analysis, a moderate correlation emerged between lyso-Gb3 in the whole AFD cohort and LVMi, as well as with GLS (coefficient for both 0.34). This observation aligns with findings from Hsu’s 2017 study, which reported that a correlation between Gb3 accumulation and LVMI was statistically significant but moderate (coefficient 0.45, *p* = 0.014) [78].

### 4.5. Study Limitations

Our study acknowledges limitations, such as the single-center and retrospective design, coupled with a relatively small sample size (considered the rarity of the condition), which may restrict the generalizability of the findings. The CMR sub-study was only preliminary, and its selection included only a minor proportion of patients based on cardiac involvement, abnormal ETT features, scheduled or ongoing therapy, and significant multiorgan involvement. Nevertheless, our observations seem consistent with existing literature, and future work aims to expand the CMR study to all consenting patients without contraindications. Additional limitations include uncertainties in variant classification and potential selection bias, as some subjects were screened through family connections. The lack of regular biomarker dosage (e.g., NT-proBNP, troponin) and inconsistent values of lyso-Gb3 further contributed to data restrictions. Precise information regarding the onset of hypertrophy was also missing for a significant proportion of patients who were diagnosed or began to be followed by us in adulthood. Moreover, the study did not quantify the impact of the non-genetic and environmental components and other comorbidities (such as HCM in the patient with D313Y), which may have influenced the findings. Lastly, the long-term effect of therapy on the occurrence of cardiac events was not fully analyzed due to these limitations, along with variability in the age of therapy initiation. Future research with larger cohorts will be essential to confirm these findings and overcome these limitations.

### 4.6. Clinical Implications and Future Directions

Our research endeavors to deepen the understanding of AFD, focusing on the clinical complexity and nuanced presentations that can affect diagnostic and therapeutic decisions, especially in females with non-classical variants. Given the high prevalence of cardiac manifestations, especially in the late-onset phenotype, also despite normal α-Gal A levels and lower lyso-Gb3 compared to men, cardiovascular screening becomes crucial in relatives and suspected cases [55]. The relationship between genetic variants and the disease is not always straightforward. A mutation might be clearly related to a disease, explaining a patient’s phenotype, but not all genetic variants are responsible for the patient’s symptomatology [32]. The pathogenicity of a variant and its relation to clinical phenotype may remain unclear, posing significant challenges to counseling, family screening, and treatment options.

The literature highlights the importance of utilizing lyso-Gb3 in blood and/or Gb3 accumulation in biopsies to supplement AFD diagnosis. Plasma lyso-Gb3 has demonstrated high sensitivity and specificity for AFD in both males and females, offering supportive diagnostic information when gene sequencing results are inconclusive. α-Gal A activity in dried blood spots has shown high sensitivity but lower specificity in males, leading to the recommendation of reflexing to gene sequencing and plasma lyso-Gb3 for disease confirmation. For females, a combination of *GLA* sequencing and plasma lyso-Gb3 provides the greatest sensitivity and specificity in diagnosis and in differentiating classic AFD from non-classic [32,50,77].

Two-thirds of all Fabry patients are genetically defined as female, emphasizing the importance of cardiologists being aware of characteristic Fabry cardiomyopathy in female patients [43], for an individualized approach to patient care, reflecting genotype, phenotype, and sex-specific factors. Accurate phenotype classification of *GLA* variants, precise differentiation between the ‘classic’ and ‘later-onset’ pathogenic phenotypes, and timely reclassification based on new supporting data can significantly benefit the clinical management of this rare and often complex disorder [7,33].

## 5. Conclusions

This study underscores the significance of sex differences in Fabry disease, evidencing phenotypic and genotypic disparities. It shows women with a generally later onset of symptoms, lesser degrees of cardiac hypertrophy, and lower impairment of GLS. In our female cohort, we observed that more severe cardiac conditions were typically correlated with classic variants, whereas VUS were often associated with more subtle disease manifestations. The early alteration of the GLS in males, which can precede the development of manifest hypertrophy, further underlines the role of subclinical damage in this multifaceted disease.

## Figures and Tables

**Figure 1 genes-14-01804-f001:**
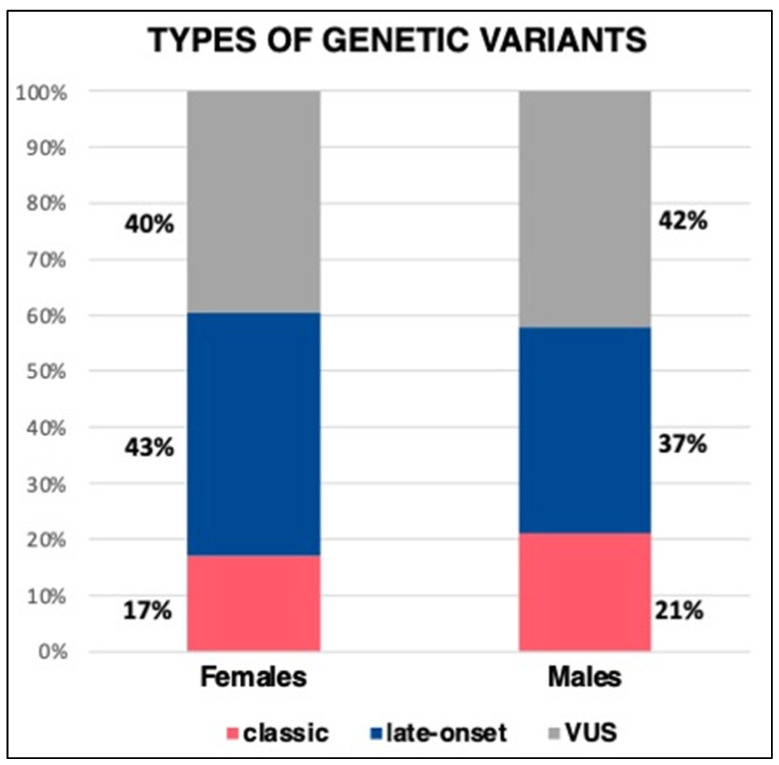
Distribution of types of variants (classic, late-onset, VUS)—difference between males and females.

**Figure 2 genes-14-01804-f002:**
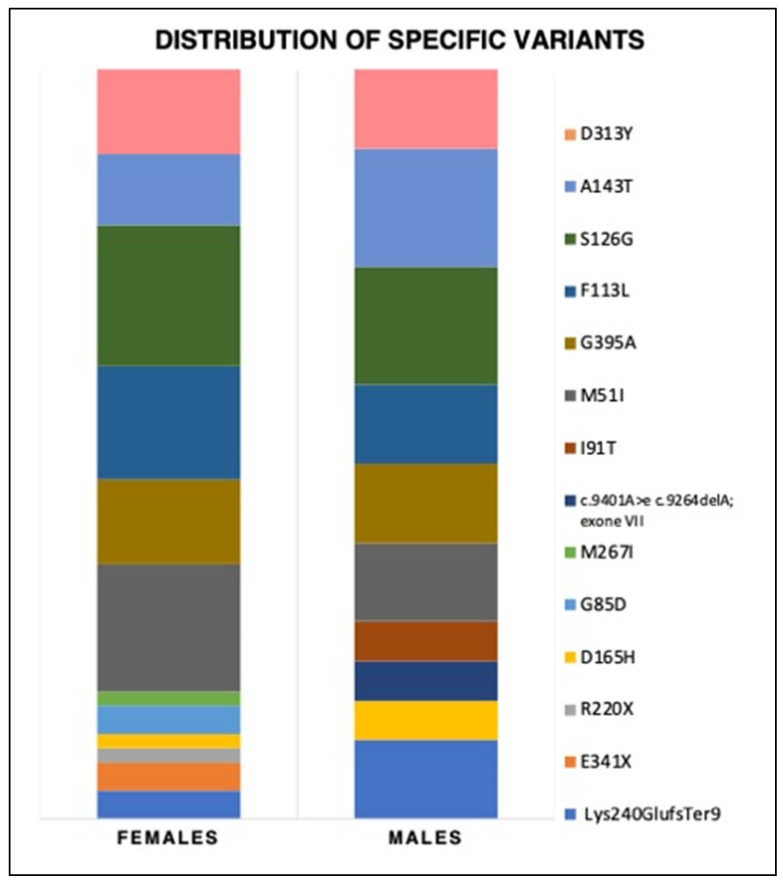
Distribution of genetic variants in detail for the single mutation found in our study population in males and females.

**Table 1 genes-14-01804-t001:** General characteristics of the patients.

	AFD Tot (N = 72)	AFD-LVH (N = 12)	AFD-N (N = 60)	NH (N = 40)
Age, yo	45 ± 16.1	60.7 ± 7.2	41.8 ± 15.6	44.1 ± 13.9
M	19 (26.4%)	3 (25%)	16 (26.7%)	26 (65%)
F	53 (73.6%)	9 (75%)	44 (73.3%)	14 (35%)
Hypertension	24 (33.3%)	8 (66.7%)	16 (26.7%)	
Diabetes	9 (12.5%)	2 (16.7%)	7 (11.7%)	
Smoke	14 (19.4%)	2 (16.7%)	12 (20%)	
Dyslipidemia	9 (12.5%)	1 (8.3%)	8 (13.3%)	
CKD	9 (12.5%)	5 (41.7%)	4 (6.7%)	
Stroke/TIA	6 (8.3%)	2 (16.7%)	4 (6.7%)	
Syncope	6 (8.3%)	2 (16.7%)	4 (6.7%)	
Fam. history SCD	1 (1.4%)	1 (8.3%)	0	
Fam. history AFD	41 (56.9%)	8 (66.7%)	33 (55%)	
Kidney transplant	6 (8%)	4 (30.8%)	2 (3.2%)	
α-Gal A (nmol/mL/h)	5.0 [2.6–9.4]	6.4 [0.3–9.6]	4.9 [2.6–9.8]	
Lyso-Gb3 (nmol/L)	1.5 [1.3–2.7]	9.2 [6.3–16.1]	1.5 [1.2–1.7]	
CV Death	2 (2.8%)	2 (16.7%)	0	
NYHA 1	53 (73.6%)	4 (33.3%)	49 (81.7%)	
NYHA 2	18 (25%)	7 (58.3%)	11 (18.3%)	
NYHA 3	1 (1.4%)	1 (8.3%)	0	
NYHA 4	0	0	0	
BP-sys (mmHg)	125 [120–135]	137.5 [126.2–143.7]	120 [115–130]	130 [120–140]
BP-dia (mmHg)	80 [70–80]	80 [71.2–80]	75 [70–80]	80 [70–80]
HR (bpm)	69.5 ± 8.8	64.7 ± 11.1	70.5 ± 8	66 ± 12.8
ERT	18 (25%)	7 (58.3%)	11 (18.3%)	
Migalastat	8 (11.1%)	1 (8.3%)	7 (11.7%)	
ICD	1 (1.4%)	1 (8.3%)	0	
PM	1 (1.4%)	1 (8.3%)	0	
*GLA* variant type:				
Classic	13 (18.1%)	5 (41.7%)	8 (13.4%)	
Late-onset	30 (41.6%)	4 (33.3%)	26 (43.3%)	
VUS	29 (40.3%)	3 (25%)	26 (43.3%)	

Where not specified, data are expressed as numbers and percentages. Abbreviations: CKD, chronic kidney disease; TIA = transitory ischemic attack; GB3, globotriaosylceramide; NYHA = New York Heart Association; BP = blood pressure; sys = systolic; dia = diastolic; *GLA*: alfa-galactosidase A gene; HR = heart rate; ERT = enzyme replacement therapy; ICD = implantable cardioverter defibrillator; PM = pacemaker; VUS = variant of uncertain significance.

**Table 2 genes-14-01804-t002:** Data related to instrumental exams.

	AFD (tot) (N = 72)	AFD-LVH (N = 12)	AFD-N (N = 60)
RBBB	4 (5.5%)	1 (8.3%)	3 (4.84%)
AVB-III	1 (1.33%)	1 (7.7%)	0
Short PR	3 (4.2%)	3 (25%)	0
D-ECG done	12 (16.7%)	8 (66.7%)	4 (6.7%)
VT/NSVT	6 (8.3%)	5 (41.7%)	1 (1.7%)
AF	7 (9.7%)	7 (58.3%)	0
PSVT	7 (9.7%)	6 (50%)	1 (1.7%)
CMR done	14 (19.4%)	6 (50%)	8 (13.3%)
LGE+ (% tot)	9 (12.5%)	5 (41.7%)	4 (6.7%)
LGE+ (% pts with CMR)	9/14 (64.2%)	5/6 (83.3%)	4/8 (50%)
N° segments LGE	2.5 [2–5]	3 [2–8.5]	2 [1.5–4.5]

Where not specified, data are expressed as numbers and percentages. Abbreviations: RBBB/LBBB = right/left bundle branch block; AVB = atrioventricular block; VT/NSVT = sustained/non-sustained ventricular tachycardia; PVC = premature ventricular complex; AF = atrial fibrillation; PSVT = paroxysmal supraventricular tachycardia; CMR = cardiac magnetic resonance; LGE = late gadolinium enhancement; SCD = sudden cardiac death.

**Table 3 genes-14-01804-t003:** Genetic variants.

Panel A	Type	Variant	tot	N° Families		Males	Females
**Pathogenic/** **likely pathogenic**	CLASSIC	Lys240GlufsTer9	4	1		2	2
		E341X	3	1		1	2
		R220X	1	1		0	1
		D165H	2	1		1	1
		G85D	4	1		1	3
		M267I	2	1		0	2
		c.9401A > e c.9264delA; exone VII	1	1		1	0
	LATE-ONSET	I91T	3	1		1	2
		**M51I**	17 (11.8%)	3		5	12
		**G395A**	12 (8.3%)	4		3	9
		**F113L**	11 (7.6%)	2		2	9
**Panel B**	**type**	**variant**	**N° tot**	**AFD-LVH**	**AFD-N**	**Males**	**Females**
Pathogenic/ likely pathogenic	CLASSIC	Lys240GlufsTer9	4	1 F	2M 1F	2	2
		E341X	2	1 F	1 F	-	2
		R220X	1	1 F	-	-	1
		D165H	2	1 M	1 F	1	1
		G85D	2	1 F	1 F	-	2
		M267I	1	-	1 F	-	1
		c.9401A > e c.9264delA; exone VII	1	-	1 M	1	-
	LATE-ONSET	I91T	1	1 M	-	1	-
		**M51I**	11 (15.3%)	1 F	10	2 (10.5%)	9 (17%)
		**G395A**	8 (11.1%)	-	8	2 (10.5%)	6 (11.3%)
		**F113L**	10 (13.9%)	1F 1M	8	2 (10.5%)	8 (15.1%)

Panel A: Distribution of *GLA* genetic variants classified per type and specific variant in the entire population who underwent a genetic examination in our clinic (173 subjects, of which 126 tested positive). Panel B: Distribution of *GLA* genetic variants classified per type and specific variant in the entire population of the study who underwent complete cardiologic clinical and imaging examination (72 patients). * Variants of uncertain significance and likely benign/benign polymorphisms are detailed in Appendix A. We grouped these three variants under a single category called “of conflicting interpretation of pathogenicity”, as reported in the ClinVar portal because their interpretation is still debated between VUS and likely benign/benign polymorphisms. We refer to https://www.ncbi.nlm.nih.gov/clinvar/, accessed on 31 July 2023, see in the text. Abbreviations: AFD-LVH, patients with left ventricle hypertrophy; AFD-N, patients without left ventricle hypertrophy; CL, classic; LO, late-onset; VUS, variant of unknown significance; F, female; M, male.

**Table 4 genes-14-01804-t004:** Comparison of echocardiographic parameters.

	AFD (tot) (N = 72)	AFD-LVH (N = 12)	AFD-N (N = 60)	NH (N = 40)	*p* Value (All Groups)	AFD-LVH vs. AFD-N	AFD-LVH vs. NH	AFD-N vs. NH
IVS (mm)	8.4 [7.0–9.2]	13 [12.1–17.3]	8 [6.5–9]	8.7 [7.1–9.4]	**<0.001**	**<0.001**	**<0.001**	0.10
PW (mm)	8.0 [7.0–9.3]	12 [10.2–14]	8 [7–9]	8.9 [7–10]	**<0.001**	**<0.001**	**<0.001**	0.05
LVMi (g/sqm)	71.5 [57.2–86.2]	116 [103.2–180.7]	67 [54–78]	76.8 [60.2–89.7]	**<0.001**	**<0.001**	**<0.001**	**0.01**
EF (%)	65 [62–67.7]	65 [62.2–69.2]	65 [61.2–67.7]	64.5 [62–66.7]	0.66	0.79	0.44	0.45
LAVi (ml/sqm)	24 [17–31]	42.5 [33.2–54]	22 [17–27]	19 [14.7–27]	**<0.001**	**<0.001**	**<0.001**	0.25
E/e’	7 [6–10]	11.5 [9.2–16.7]	7 [5.6–9]	7 [6–8]	**0.001**	**0.001**	**<0.001**	0.71
TR-Vmax (m/s)	2.3 [1.9–2.4]	2.3 [2.1–2.4]	2.2 [1.8–2.4]	2.1 [1.9–2.3]	0.28	0.16	0.11	0.83
LV-GLS (-%)	−18 [−16/−21]	−11.5 [−9.2/−16.7]	−19 [−17/−22]	−20 [−18/−21]	**<0.001**	**<0.001**	**<0.001**	0.24
LV-MD (ms)	39 [29–60]	74 [59.2–90]	36 [29–46]	31 [27–41]	**<0.001**	**<0.001**	**<0.001**	0.13
Papill Hypertr.	4 (5.6%)	1 (8.3%)	3 (5%)	0				

Data are expressed as median [IQR] and n° (%) as appropriate. Abbreviations: IVS, interventricular septum; PW, posterior wall; LVMi, LV mass index; EF, ejection fraction; LAVi, left atrial volume index; E/e’, ratio between E wave of mitral flow and e’ at Tissue Doppler; TR-V, tricuspid regurgitation velocity; LV-GLS, left ventricle global longitudinal strain; MD, mechanical dispersion.

**Table 5 genes-14-01804-t005:** Sex differences: comparison between females and males in the total AFD population of the study (n = 72).

	AFD Tot F (N = 53)	AFD Tot M (N = 19)	*p*-Value
AGE, yo	47.2 ± 16.2	38.8 ± 14.6	**0.046**
CLASSIC variant	9 (17%)	4 (21%)	0.73
LATE-ONSET	23 (43.4%)	7 (36.9%)	0.78
VUS	21 (39.6%)	8 (42.1%)	1
α-Gal A (nmol/mL/h)	7.9 [4.6–11.3]	1.8 [0.3–3.35]	**<0.001**
LYSO-GB3 (nmol/L)	1.5 [1.1–1.7]	1.9 [1.5–17.3]	**0.02**
ERT	14 (26.4%)	4 (21.1%)	0.76
Migalastat	6 (11.3%)	2 (10.5%)	1
V-AR	4 (7.5%)	2 (10.5%)	0.65
AF	5 (9.4%)	2 (10.5%)	1
Pts with LVH	9 (17%)	3 (15.8%)	1
LVMi (g/sqm)	66 [54–85.5]	78 [73–96]	**0.03**
EF, (%)	65 [62–67.5]	64 [60–68]	0.38
E/e’	8 [6–10]	7 [5–10]	0.22
LV-GLS (-%)	−19 [−17, −22]	−17 [−14, −20]	**0.02**
MD (ms)	37 [29.2–49.7]	41 [29–60]	0.65
LGE+	4 [57% of CRM]	7 [87.5%]	**0.001**
LGE segments, n°	3 [1–3]	2 [2–5]	**0.003**

Where not specified, values are expressed as numbers and percentages. Abbreviations: VUS, variant of unknown significance; ERT, enzyme replacement therapy; V-AR, ventricular arrhythmias; AF, atrial fibrillation; LVH, left ventricular hypertrophy; LVMi, left ventricle mass index; EF, ejection fraction; E/e’, ratio between E wave of mitral flow and e’ at Tissue Doppler; LV-GLS, left ventricle global longitudinal strain; MD, mechanical dispersion. LGE, late gadolinium enhancement.

**Table 6 genes-14-01804-t006:** Sex differences in AFD-LVH patients (on the left) and in AFD-N patients (on the right).

	AFD-LVH F (N = 9)	AFD-LVH M (N = 3)	*p* Value	AFD-N F (N = 44)	AFD-N M (N = 16)	*p* Value
AGE, yo	62.5 ± 3.9	55.3 ± 13	0.44	44 ± 16	35.7 ± 12.9	**0.048**
CLASSIC MUTATION	4 (40%)	1 (33.3%)	1	5 (11.4%)	3 (18.7%)	0.43
LATE-ONSET	2 (20%)	2 (66.7%)	0.24	21 (47.7%)	5 (31.3%)	0.38
VUS	3 (30%)	0	0.50	18 (40.9%)	6 (37.5%)	1
α-Gal A (nmol/mL/h)	8.5 [4.8–10.9]	0.3 [0–0.3]	**0.017**	6.8 [4.5–13.8]	2.75 [0.57–4.05]	**0.003**
LYSO-GB3 (nmol/L)	8.5 [2.8–9.3]	22.9 [9.4–22.9]	0.13	1.5 [1.1–1.7]	1.65 [1.4–2.7]	**0.04**
ERT	5 (55.6%)	2 (66.7%)	1	9 (20.5%)	2 (12.5%)	0.71
Migalastat	0	1	0.25	6 (13.6%)	1 (6.3%)	0.66
V-AR	3 (33.3%)	2 (66.7%)	0.52	1 (2.3%)	0	1
AF	5 (55.6%)	2 (66.7%)	1	0	0	1
LVMi (g/sqm)	115 [102–140]	227 [116–227]	0.06	61.5 [53–74.2]	77.5 [69.2–80]	**0.008**
EF (%)	65 [62–72]	65 [62–65]	0.86	65.5 [62–67.25]	64 [60–68.75]	0.42
E/e’	13 [7.5–15]	10 [7.7–17]	0.72	7 [6–9]	6.5 [5–7.75]	0.09
LV-GLS (-%)	−12.6 [−10/−18.5]	−10 [−9/−10]	0.20	−20 [−17/−22]	−17 [−16/−20]	**0.01**
MD (ms)	70 [48–90.5]	87 [57–87]	0.72	35 [29–45]	38 [28.2–58.2]	0.78

Where not specified, values are expressed as numbers and percentages. Abbreviations: VUS, variant of unknown significance; ERT, enzyme replacement therapy; V-AR, ventricular arrhythmias; AF, atrial fibrillation; LVH, left ventricular hypertrophy; LVMi, left ventricle mass index; EF, ejection fraction; E/e’, ratio between E wave of mitral flow and e’ at Tissue Doppler; LV-GLS, left ventricle global longitudinal strain; MD, mechanical dispersion. LGE, late gadolinium enhancement.

**Table 7 genes-14-01804-t007:** Comparison between females with LVH and without.

	AFD-LVH Females (N = 9)	AFD-N Females (N = 44)	*p* Value
**Age (yo)**	62.5 ± 3.9	44 ± 16	**0.001**
**α-Gal A (nmol/mL/h)**	8.5 [4.8–10.9]	6.8 [4.5–13.8]	0.85
**Lyso-Gb3 (nmol/L)**	8.5 [2.8–9.3]	1.5 [1.1–1.7]	**0.048**
**Classic mutation**	4 (40%)	5 (11.4%)	**0.03**
**Late-onset**	2 (20%)	21 (47.7%)	0.27
**VUS**	3 (30%)	18 (40.9%)	1
**V-AR**	3 (33.3%)	1 (2.3%)	**0.01**
**AF**	5 (55.6%)	0	**<0.001**
**LVMi (g/sqm)**	115 [102–140]	61.5 [53–74.2]	**<0.001**
**EF (%)**	65 [62–72]	65.5 [62–67.25]	0.88
**E/e’**	13 [7.5–15]	7 [6–9]	**0.009**
**GLS (-%)**	−12.6 [−10/−18.5]	−20 [−17/−22]	**0.001**
**MD (ms)**	70 [48–90.5]	35 [29–45]	**0.002**

Abbreviations: VUS, variant of unknown significance; ERT, enzyme replacement therapy; V-AR, ventricular arrhythmias; AF, atrial fibrillation; LVH, left ventricular hypertrophy; LVMi, left ventricle mass index; EF, ejection fraction; E/e’, ratio between E wave of mitral flow and e’ at Tissue Doppler; LV-GLS, left ventricle global longitudinal strain; MD, mechanical dispersion. LGE, late gadolinium enhancement.

**Table 8 genes-14-01804-t008:** Comparison between AFD-N and control groups in both females and males separately.

	AFD-N Females N = 44	NH Females N = 14	*p* Value	AFD-N Males N = 16	NH Males N = 26	*p* Value
Age, yo	44 ± 16	48.7 ± 11.5	0.30	35.7 ± 12.9	41.6 ±14.6	0.20
LVMi (g/sqm)	61.5 [53–74.2]	64 [50.5–78.5]	0.89	77.5 [69.2–80]	83 [72–81.7]	0.11
EF (%)	65.5 [62–67.25]	65 [62.7–66.25]	0.66	64 [60–68.7]	63.5 [60.7–67.2]	0.87
LAVi (ml/sqm)	22 [17–27]	25.7 [17.7–31.2]	0.35	24 [17–26]	17.5 [14–19.7]	0.14
E/e’	7 [6–9]	7.15 [5.9–8.3]	0.52	6.5 [5–7.7]	7 [5.9–8.1]	0.29
TR-Vmax (m/s)	2.2 [1.8–2.4]	2.2 [2–2.4]	0.72	2 [1.8–2.4]	2 [1.9–2.3]	0.84
LV-GLS (-%)	−20 [−17/−22]	−20 [−16.8/−21.5]	0.71	**−17 [−16/−20]**	**−20 [−19/−21.2]**	**0.004**
LV-MD (ms)	35 [29–45]	29.6 [24.5–38.3]	0.08	38 [28.2–58.2]	34 [27.5–42.5]	0.50

Abbreviations: LVMi, left ventricle mass index; EF, ejection fraction; E/e’, ratio between E wave of mitral flow and e’ at Tissue Doppler; TRV, tricuspid regurgitation velocity; LV-GLS, left ventricle global longitudinal strain; MD, mechanical dispersion.

## Data Availability

The data presented in this study are available on request from the corresponding author.

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
