# Peer review of "Sex Differences in Anderson–Fabry Cardiomyopathy: Clinical, Genetic, and Imaging Analysis in Women"

_genes, 2023, doi:10.3390/genes14091804_

Round 1

Reviewer 1 Report

Dear authors,

First of all, I would like to congratulate the authors for a very interesting and well presented work.

In this work, authors investigated clinical and genetic characteristics of a group of Fabry disease patients. Clinical characteristics are well documented and well explained. I believe this work can help to better understand the cardiac manifestation differences in different groups of Fabry patients.

My main concern on this manuscript is lack of analysis between treated vs un-treated patients. For example, 18 out of 72 AFD Tot patients were on ERT and 8 were on Migalastat. Authors should include a section specific to treatment summarizing the parameters in ERT treated vs Migalastat treated vs un-treated patients.

Authors highlighted ‘inconsistent values of Lyso-GB3’ in the discussion section. I would be happy to see a specific section on Lyso-GB3 with detailed analysis of each group, which may explain those inconsistencies. Of note, ERT and Migalastat can have an effect on Lyso-GB3. And this should be discussed.

Finally, there is no analysis of pre and post ERT&Migalastat values. I wonder if any of the patients were analyzed pre-treatment (with ERT or Migalastat) and further analyzed post-therapy?

I would like to make two small formatting suggestions:

Can you make sure the fonts and font sizes are coherent throughout the manuscript?

Can you make sure resolution of figures are high quality?

Some level of English language editing is needed.

Reviewer 2 Report

The authors presented a nice and well-written original manuscript entitled "Sex differences in Anderson-Fabry Cardiomyopathy: clinical, genetic, and imaging analysis in Women". This interesting retrospective observational study presents key lab and genetic findings and clinical, echocardiographic and cardiac imaging features in different patients with Fabry's disease. It is of note the high quality of the evidence showing correlation of late-onset phenotype with hypertrophic cardiomyopathy and high levels of lyso-GB3 detected. The graphical abstract presents the key information for the reader, who can immediately detect the quality of the study as well as its importance for clinical practice. Figure 2 is almost brilliant and summarizes the profile of the main genetic variants in male and female population. I have some suggestions and comments about some topics in the manuscript: 

1. A minor point to be evaluated by the authors is the presentation of the GLA gene in their manuscript, which should be always presented in italics

2. There is no formal need to present in the third paragraph of the Introduction a specific text related to the ACMG criteria to classify genetic variants, as this journal's readers are mainly composed by geneticists, physicians, neurologists and researchers in rare diseases which are nowadays used to the classification. 

3. In Table 3, I think there is no need to present the likely benign and benign variants in Panel A and B. One suggestion, for example, is to present the information in a supplementary content. 

4. Have the authors any data related to the occurrence of clinically significant dysautonomia in their sample? 

5. Another suggestion is the inclusion, for example, of a Table comparing some of the authors results in this study with the classic results related to cardiac involvement observed in other cohorts, like in the Fabry Outcome Survey. 
